

# A physical fitness–evaluation system for outstanding Chinese male boxers

Guodong Wu[1], Yuqiang Guo[2], Liqin Zhang[3] and Chao Chen[2]

[1] Jiangsu Vocational Institute of Commerce, Nanjing, China
[2] School of Athletic Performance, Shanghai University of Sport, Shanghai, China
[3] Inner Mongolia Institute of Sports Science, Inner Mongolia, China

## ABSTRACT

**Background:** We sought to create a system to evaluate the physical fitness of outstanding Chinese male boxers that included an evaluation index, fitness level criteria, and modeling. This system was then used to assess athletes' physical fitness and development.

**Methods:** Documentation, expert interviews, questionnaires, measurements, and statistical analyses were used in this study.

**Results:** The physical fitness evaluation system included the following three components: (1) body shape indexes ($n = 4$) including the backhand upper arm circumference differential, finger span height, Cottrell index, and pelvic width/shoulder width $\times$ 100; (2) body function indexes ($n = 4$) including relative maximum anaerobic power, relative maximal oxygen uptake, and creatine kinase and testosterone concentrations; and (3) athletic quality indexes ($n = 9$) including the speed strength index, the backhand straight punch strength, 3-min cumulative punching force, backhand straight punch reaction time, backhand straight punch speed, 30-m sprint, 9-min double shake jump rope, 1-min double shake jump rope, and sitting forward bend tests. A five-point grading system to evaluate physical fitness was established and an evaluation model was proposed.

**Conclusions:** The reference values were determined to be objective and effective using a back substitution process. Individual and differential assessments reflected the athletes' level of physical fitness. The critical values were established under the best and worst conditions and the optimal values were found to be valid and effective.

## INTRODUCTION

In 2013 a major change was made in how a boxing contest could be won. The International Boxing Federation replaced the "point-to-win" system with the "10-point must" system. Under the new rules each round is scored using four levels: 10–9, 10–8, 10–7, and 10–6. Judges now decide who wins or loses based on four criteria: the number of quality punches landed in the target area, the domination of the bout by technical and tactical superiority, competitiveness, and infringement of the rules (*Davis et al., 2018*). The "10-point must" system has propelled boxing into the same realm as combat fighting. To gain competitive advantages, boxers must be more proactive and aggressive with their punches and effectively strike their opponents to demonstrate their superior athletic strength

Corresponding author
Chao Chen,
chenchao2019@sus.edu.cn

(*Bianco et al., 2013*). These requirements place a high demand on the boxer's athletic ability to sustain three rounds and necessitates great strength and physical fitness (*Davis et al., 2018*). One study found that in the men's Olympic boxing competition (3 × 3 min) approximately 1.3 actions occurred per minute, including ~20 punches, ~2.5 defensive movements, and ~47 vertical hip movements during three rounds that lasted up to 252 s. Comprehensive physical fitness is the foundation of a boxer's physical conditioning. This foundation emphasizes training in specific areas including strength, speed, endurance, limb coordination, and agility. This foundation ensures the smooth implementation of technical and tactical skills in competition (*Miarka et al., 2011*).

However, technical and tactical advantages cannot be solely relied upon to win in combat-based programs; athletes possessing these skills are unable to beat opponents at similar levels if they lack adequate physical fitness (*Chaabene et al., 2015*; *Coswig et al., 2018*). Physical fitness training is essential for athletes to improve their overall fitness levels. Outstanding boxers must excel in all aspects of physical fitness to avoid obvious weaknesses during combat (*Davis, Wittekind & Beneke, 2013*). Boxing is a combination of offensive and defensive actions. It is a high-intensity and energy-demanding sport with intermittent rounds of competition (*Kamandulis et al., 2018*; *Loturco et al., 2018*) and an athlete in competition uses primarily anaerobic energy supplemented by aerobic energy. Physical fitness training not only provides a solid foundation for executing the challenging combination of offensive and defensive moves but also lifts an athlete's fighting spirit and serves as a preventive measure against injury (*Zhou et al., 2022*). Evaluating an athlete's physical fitness is important for determining whether the training load is appropriate. An appropriate training program can avoid prolonged training fatigue and injury. Simultaneously, the physical fitness data-collection process can help coaches better understand their athletes and identify individual problems at the current stage of training. Coaches can adjust their training plans to tailor the training process and make it more thorough and meticulous (*Halson, 2014*; *Woods et al., 2018*).

Due to various restrictions, existing research on the physical fitness evaluation index for boxers is rather limited (*Ambrozy et al., 2021*; *Han et al., 2020*; *Loturco et al., 2021*; *Woods et al., 2018*). One study found that being heavier, taller, and having longer arms and thicker bodies but similar body mass index and waist circumference seemed to be important indicators that distinguished champions from challengers in world championships (*Han et al., 2020*). *Merlo et al. (2023)* used unsupervised machine learning algorithms and found that the strength level of the upper and lower limbs of young boxers is an important factor affecting the velocity of straight boxing punches and the impact force of the rear hand punch. Identifying these physical characteristics helps in evaluating young boxers. At the same time, *Loturco et al. (2021)* studied the impact of how improvements to the upper and lower limb strength of boxers effected their punch performance, using national boxers as the research subjects. Their results also showed that upper and lower limb strength improvement is directly related to how well boxers can punch. Since agility is a key factor that affects the performance of boxers in competitions, *Zhong & Bu (2022)* developed a structural model for evaluating the agility of female boxers. After the model was verified, it proved that this model could be used for evaluating the agility of female boxers. It is

necessary to conduct a comprehensive study that integrates various fitness indexes. By creating a physical fitness evaluation index, athletes' level of fitness can be evaluated quantitatively and their strengths and weaknesses can be accurately assessed. This allows for active control of the training process and targeted improvements to an athlete's fitness can be made. It can also provide scientific and clear guidance for physical fitness training. This study aimed to create a physical fitness evaluation system to assess the physical development of outstanding male boxers in China.

## METHODS

### Subjects

Chinese male boxers were recruited at two stages during this research. During the first stage, 149 participants were recruited including 76 elite and 73 first-class athletes. During the second stage, 25 participants were recruited including eight elite athletes and 17 first-class athletes. The participants were divided into lightweight, middleweight, and heavyweight categories according to International Boxing Federation standards. Participation was voluntary. This study was approved by the Shanghai University of Sport Research Ethics Committee (approval no. 102772021RT102). This study was performed in accordance with the Helsinki Declaration. We received written informed consent from all participants. Table 1 presents basic participant information.

### Procedures

#### Research design

Testing was performed in two stages from December 2018 through January 2020 and January 13–16th, 2020 due to the large number of testing indexes used and the number of participants. Prior to formal testing, 20 male boxers from Hubei Province were selected to pretest the testing instruments and process. Formal testing was performed by coaches, research coaches, and fitness instructors. Body shape indexes were measured using a height gauge, weight gauge, ruler, soft ruler, bend angle measuring tool, and a body composition analyser (InBody 3.0; InBody, Seoul, Korea). Body function indexes were measured using a stopwatch, field-marking poles, whistles, a MONARK837 power bicycle (Monark, Varberg, Sweden), and a lactate analyser (Lactate Scout 4; EKF Diagnostics Inc., Cardiff, UK). Athletic quality indexes were measured using weight plates, barbells, a 3-kg solid ball, a tape measure, a neck-weighting hat, rope, a sit-and-reach tester, a StrikeTec combat data-collection system (StrikeTec-Boxing Performance Tracking, Dallas, TX, USA), and sandbags (Jiuri Mountain, Fujian, China).

A variety of databases, including the Web of Science, EBSCOHost, and SPORTDiscus were searched and relevant articles published over the last 10 years were retrieved and reviewed. A close examination of recent articles regarding boxing policy was conducted to establish a theoretical foundation for the design of boxing evaluation and diagnostic systems. Thirteen experts in the field were interviewed face-to-face or *via* email or telephone. The interviews were conducted to gain insight into typical physical fitness evaluation indexes, methods for selecting such indexes, steps for building an assessment

**Table 1 Basic participant information.**

| Testing phase | Group | Sample (*n*) | Age (yrs) | Training age (yrs) |
|---|---|---|---|---|
| The first stage | Lightweight | 41 | 21.46 ± 4.15 | 7.22 ± 4.08 |
| | Middleweight | 68 | 22.71 ± 4.51 | 8.12 ± 3.66 |
| | Heavyweight | 40 | 23.20 ± 4.83 | 7.28 ± 3.63 |
| The second stage | Lightweight | 6 | 19.17 ± 0.98 | 5.67 ± 1.21 |
| | Middleweight | 14 | 21.60 ± 3.81 | 7.60 ± 3.81 |
| | Heavyweight | 5 | 24.00 ± 8.03 | 9.60 ± 6.95 |

Note:
Training age, the length of time that systematic training is maintained.

system, and details of fitness testing. Expert feedback and opinions were compiled to provide evidence of the feasibility of the study.

The following two research surveys were also conducted: the Unk "Survey on Physical Fitness evaluation Indexes for Excellent Male Boxers" and the "Expert Questionnaire on Weighting Coefficients of Physical Fitness evaluation Indexes for Excellent Male Boxers".

The physical fitness evaluation index system was created using the following steps: (1) the initial screening of Chinese male boxers using fitness indexes; (2) the compilation of a preliminary index; (3) the distribution of a Delphi survey among selected field experts to analyse their activeness and authority in the field and to determine the coordination coefficient. (Factor analysis was used to eliminate unfit indexes); and (4) the establishment of a testing index with weight coefficients.

A three-round Delphi survey of 20 university professors, senior level experts, and boxing coaches was conducted and structural validation testing was performed. A five-point Likert scale was adopted with one point meaning "not important," two points meaning "somewhat unimportant," three points meaning "average," four points meaning "somewhat important," and five points meaning "important." The response rates of each round of the survey were 100%, 100%, and 95%, respectively.

### Determination of the weighting system

Factor analysis (principal component analysis) was used to analyse the test data and the eigenvalues of each factor in the first-level indicators and the contribution rate of each factor were obtained. By dividing the contribution rate of each factor by the sum of the cumulative contribution rate of factors, the weight of each factor could be calculated (Formula (1)). Since a typical evaluation index is selected for each factor, the weighted factors of the typical evaluation index for the first-level indicators can be obtained using the following (Formula (1)):

where $t_i$ is the weighted factor of a certain indicator under a given factor, $b_i$ is the contribution rate of the indicator

$$t_i = \frac{b_i}{\sum_{j=1}^{k} b_y} \tag{1}$$

under the given factor, i represents a certain indicator or the code of an indicator, and $\sum_{j=1}^{k} b_y$ is the cumulative contribution rate of all the indicators under this factor. Also, k represents the number of indicators used for factor analysis.

The weighted typical evaluation index of each factor in the first-level indicators obtained from Formula (1) was used along with the corresponding individual indicator scoring criteria to calculate the total score of the first-level indicators. This allowed for the data to be standardized. Factor analysis (principal component analysis) was performed on the standardized data to obtain the characteristic values and contribution rates of each factor. The weight of each factor in the evaluation index system could be calculated using Formula (1). By multiplying the weight of each first-level indicator in each factor with the weighted index of that factor and then adding them together (Formula (2)), the weighted index of the first and second-level indicators was obtained. Finally, the weighted model of the physical fitness evaluation index for elite male boxers was obtained by converting the weighted values of the first, second, and individual indicators, using the following (Formula (2)):

$$Ti = \sum_{j=1}^{k} (mij \times nij) \tag{2}$$

where $T_i$ is the weighted factor of a certain indicator, $m_{ij}$ is the weighted index of a certain factor, and $n_{ij}$ is the weighted factor of an indicator.

### Evaluation criteria

To accurately assess the physical fitness levels of Chinese male boxers it was important to establish a structured assessment standard that reflects an athlete's physical fitness level. The specific steps we took were as follows: (1) the percentile method and the 20-point scoring method were adopted to establish individual scoring criteria for each indicator for all boxers under the three weight groups; (2) the weighted score was calculated based on the weighted factor of each indicator and the total score of a first-level indicator was calculated by adding the weighted scores of all indicators; (3) the score of each first-level indicator was multiplied by its corresponding weighted factor to obtain a weighted score and the weighted scores of all first-level indicators were added together to obtain a comprehensive score; and (4) the final score was converted to a percentile and a five-level assessment method (*i.e.*, poor, fair, average, good, excellent) was used to evaluate the physical fitness levels of the boxers.

### Statistical analysis

Data analysis was performed using SPSS 24.0 (IBM Corp., Armonk, NY, USA) and the significance level was set at $p < 0.05$. The normality of the fitness test data was determined using the Kolmogorov–Smirnov test. Factor analysis was used to select the indicators. Kaiser orthogonal rotation is primarily used for rotating the factor loading matrix in factor analysis to facilitate interpretation of factors. It generally requires that the factor loadings be no less than 0.6 and requires that cross-loadings do not exceed 0.4, with unsuitable

indicators being removed. Kaiser–Meyer–Olkin (KMO) and Bartlett's test were used to check whether the data was suitable for factor analysis. Typically, with KMO > 0.5 and $p < 0.01$ it indicates a correlation among variables, suggesting that the factor analysis is appropriate (*Clark et al., 2019*; *Korucu et al., 2019*). Factors with eigenvalues >1 were removed and a cumulative contribution rate >60% was achieved (*Huang et al., 2021*). A combination of the percentile method and the 20-points scoring method was used to formulate the evaluation criteria and multiple chi-squared tests were performed for regression analysis to validate the evaluation criteria.

## RESULTS

### Construction of a physical evaluation index system

Representative indicators were selected to form a preliminary system. These indicators were further refined through expert interviews resulting in a final index system consisting of three primary indicators, 20 secondary indicators, and 81 tertiary indicators. The results of the Delphi surveys conducted with experts were also analysed. The authoritative coefficient (Cr) was 0.877, surpassing the threshold level of 0.7, which indicates a high level of confidence. The authoritative coefficient was used to determine the expert's level of authority in the subject matter. It measures the degree of familiarity the expert has with the subject matter by using certain indicators. Kendall's W indicates the level of agreement among experts regarding the evaluated items; its coefficients were 0.673 ($p < 0.001$) and 0.678 ($p < 0.001$) for the second and third survey rounds, respectively. This indicated a high level of agreement among experts. Indicators were selected based upon two coefficients: Mj, the level of agreement with the opinion and Vj, the amount the opinions varied, with the criteria that Mj ≥ 4.0 and Vj ≤ 0.25 (Table 2). High Mj and low Vj values usually suggest a high approval rating for an indicator. Mj represents the mean value of each indicator score, which represents the degree to which the opinions agree. Vj represents the coefficient of the varying opinions for each indicator score, which indicates the degree to which opinions varied. The higher the Mj value and the lower the Vj value, the greater the experts' opinions agreed on the indicators (*Rodríguez-Mañas et al., 2013*; *Weir et al., 2015*; *Kroshus et al., 2019*; *Huang et al., 2021*).

Statistical analysis was conducted on data from 49 physical fitness indicators obtained from 149 male boxing athletes following the Delphi screening. First, factor load analysis was performed for the different indicators of physical form and the results showed that waist circumference (0.443), height (0.277), and weight (0.388) were below the high-load factor (load > 0.6). These factors were eliminated based on expert advice. The Kaiser-Meyer-Olkin (KMO) score for the physical form indicators was 0.626 with a significance level of $p < 0.001$. This met the criteria for factor analysis. Four factors with eigenvalues >1 contributed to an accumulated contribution rate of 82.4%, exceeding the recommended value of 60%. The four factors represented key information from the physical form indicators and reflected the basic physical form of athletes in different weight categories. The original matrix was subjected to Kaiser orthogonal rotation (Table 3) and the physical form indicators were the circumference factor, length factor, plumpness factor, and width factor. These factors were based on the experts' advice. Factors that were closely related to

**Table 2 Primary indexes significance score and coefficient of variation.**

| Primary indexes | Testing indexes | CV (Vj) | Significance score (Mj ± Sj) | Testing indexes | CV (Vj) | Significance score (Mj ± Sj) |
|---|---|---|---|---|---|---|
| Body shape | Height | 0.097 | 4.72 ± 0.46 | Finger span height | 0.146 | 4.48 ± 0.65 |
| | Pelvic width/shoulder width × 100 | 0.102 | 4.68 ± 0.48 | Neck circumference | 0.174 | 4.36 ± 0.76 |
| | Chest circumference | 0.185 | 4.28 ± 0.79 | Waist circumference | 0.147 | 4.40 ± 0.65 |
| | Forehand upper arm circumference differential | 0.184 | 4.24 ± 0.78 | Backhand upper arm circumference differential | 0.111 | 4.56 ± 0.51 |
| | Weight | 0.102 | 4.68 ± 0.48 | Body fat percentage | 0.184 | 4.24 ± 0.78 |
| | Cottrell index | 0.161 | 4.36 ± 0.70 | | | |
| Body function | Relative average anaerobic power | 0.131 | 4.48 ± 0.59 | Relative maximum anaerobic power | 0.106 | 4.64 ± 0.49 |
| | Relative VO$_{2max}$ | 0.097 | 4.72 ± 0.46 | Hemoglobin | 0.119 | 4.68 ± 0.56 |
| | Hematocrit | 0.131 | 4.48 ± 0.59 | RBCs | 0.097 | 4.72 ± 0.46 |
| | Blood urea | 0.147 | 4.40 ± 0.65 | Creatine kinase | 0.106 | 4.64 ± 0.49 |
| | BLA | 0.161 | 4.36 ± 0.70 | Testosterone | 0.106 | 4.64 ± 0.49 |
| | Cortisol | 0.173 | 4.32 ± 0.75 | HR$_{max}$ | 0.131 | 4.48 ± 0.59 |
| Athletic quality | Squat | 0.161 | 4.40 ± 0.71 | Bench press | 0.097 | 4.72 ± 0.46 |
| | Forehand straight punch force | 0.128 | 4.56 ± 0.58 | Backhand straight punch force | 0.097 | 4.72 ± 0.46 |
| | Medicine ball | 0.111 | 4.56 ± 0.51 | Standing long jump | 0.109 | 4.60 ± 0.50 |
| | Speed strength index | 0.102 | 4.68 ± 0.48 | Maximal pull-ups | 0.097 | 4.72 ± 0.46 |
| | 1-min barbell push | 0.102 | 4.68 ± 0.48 | 3-min cumulative punching force | 0.092 | 4.76 ± 0.44 |
| | Anterior cervical muscle load limit flexion and extension | 0.125 | 4.56 ± 0.58 | Back neck muscle load limit flexion and extension | 0.102 | 4.68 ± 0.48 |
| | Forehand straight punch reaction time | 0.092 | 4.76 ± 0.44 | Backhand straight punch reaction time | 0.097 | 4.72 ± 0.46 |
| | Forehand straight punch speed | 0.106 | 4.64 ± 0.49 | Backhand straight punch speed | 0.109 | 4.60 ± 0.50 |
| | 30-m sprint | 0.174 | 4.36 ± 0.76 | 100-m sprint | 0.131 | 4.48 ± 0.59 |
| | 400-m sprint | 0.106 | 4.64 ± 0.49 | 12-min sprint | 0.109 | 4.60 ± 0.50 |
| | 9-min double shake jump rope | 0.097 | 4.72 ± 0.46 | 1-min double shake jump rope | 0.106 | 4.64 ± 0.49 |
| | Eight-way fast slide in boxing ring | 0.102 | 4.68 ± 0.48 | 1-min hexagon jump | 0.111 | 4.56 ± 0.51 |
| | Sitting forward bend | 0.113 | 4.52 ± 0.51 | Shoulder flexibility | 0.111 | 4.56 ± 0.51 |

Note:
The coefficient of variation (Vj) was used indicating the degree of difference among experts' understanding of an indicator's relative importance. The survey design used a five-point Likert scale (one point, not unimportant; two points, somewhat unimportant; three points, average; four points, somewhat important; five points, important). $M_j$ represents the arithmetic mean of different expert scores, and $S_j$ represents the standard deviation of the scores. $V_j = M_j/S_j$. VO$_{2max}$, maximal oxygen uptake; RBCs, red blood cells; BLA, blood lactate; HR$_{max}$, maximum heart rate.

boxing, *i.e.*, "often used during actual training," "easy to operate," and "highly sensitive," were selected as representative factors from each upper-level group. Four indicators (backhand upper arm circumference differential, finger span height, Cottrell index, and pelvic width/shoulder width × 100) were considered tertiary indicators of physical form. Factor analysis was performed on the 12 body function indicators. Among these factors, the red blood cell hematocrit factor load (0.497) was lower than the high-load factor standard and was therefore eliminated. The KMO and Bartlett's test results revealed a

**Table 3 Rotated factor loading matrix after rotation of body shape.**

| Index | 1 | 2 | 3 | 4 |
|---|---|---|---|---|
| Backhand upper arm circumference differential | 0.963 | | | |
| Forehand upper arm circumference differential | 0.959 | | | |
| Neck circumference | 0.806 | | | |
| Waist circumference | 0.775 | | | |
| Finger span height | | 0.929 | | |
| Cottrell index | | | 0.872 | |
| Body fat percentage | | | 0.723 | |
| Pelvic width/shoulder width × 100 | | | | 0.684 |

KMO value of 0.659, with a significance level of $p < 0.001$. This met the factor analysis criteria.

The original matrix was subjected to Kaiser orthogonal rotation (Table 4) and four factors (anaerobic ability, aerobic ability, recovery ability, and hormone levels) with eigenvalues >1 contributed to a cumulative contribution rate of 72.9%. Relative maximum anaerobic power, relative maximal oxygen uptake ($VO_{2max}$), creatine kinase, and testosterone were considered tertiary indicators of physical function based on the results of the Kaiser–Varimax rotation.

As athletic ability is an essential skill for boxers (*Walilko, Viano & Bir, 2005*) statistically significant selection of the factors was made using the key components of athletic quality: strength, speed, endurance, coordination, agility, and flexibility. A factor load analysis was performed on each set of indicators. All indicators with loads >0.6 were included. The 1-min barbell push, which had a load value of 0.465, did not qualify. KMO and Bartlett's tests showed that all five sets of athletic indicators met the criteria of KMO > 0.5 and $p < 0.01$. Principal component analysis identified three factors with eigenvalues >1: speed strength, maximum strength, and strength endurance. Three speed factors had eigenvalues >1: reaction speed factor, action speed factor, and movement speed factor. One factor qualified in each of the endurance, coordination and agility, and flexibility groups when using the Kaiser–Guttman rule. These factors were then designated as the endurance factor, coordination and agility factor, and flexibility factor, respectively. All sets of factors met the requirement of a cumulative contribution rate >60%. Kaiser–Varimax rotation was performed on the original matrix of the five sets of indicators (Table 5) to identify nine tertiary indicators of athletic ability, including speed strength index (strength), backhand straight punch force (strength), 3-min cumulative punching force (strength), backhand straight punch reaction time (speed), backhand straight punch speed (speed), 30-m sprint (speed), 9-min double shake jump rope (endurance), 1-min double shake jump rope (coordination and agility), and sitting forward bend (flexibility).

Finally, a weighted factor model to evaluate the physical fitness of the boxers was developed through the conversion of the first, second, and weighted coefficients of each individual indicator (Table 6). Factor analysis was performed on seven first-level indicators

**Table 4 Rotated factor loading matrix after rotation of body function.**

| Index | 1 | 2 | 3 | 4 |
|---|---|---|---|---|
| Relative maximum anaerobic power | 0.974 | | | |
| Relative average anaerobic power | 0.960 | | | |
| BLA | 0.894 | | | |
| Relative $VO_{2max}$ | | 0.959 | | |
| Hemoglobin | | 0.930 | | |
| RBCs | | 0.861 | | |
| $HR_{max}$ | | 0.818 | | |
| Creatine kinase | | | 0.877 | |
| Blood urea | | | 0.703 | |
| Testosterone | | | | 0.932 |
| Cortisol | | | | 0.849 |

**Table 5 Rotated factor loading matrix after rotation of body function.**

| Group | Index | 1 | 2 | 3 |
|---|---|---|---|---|
| Strength | Speed strength index | 0.915 | | |
| | Standing long jump | 0.698 | | |
| | Medicine ball | 0.608 | | |
| | Backhand straight punch force | | 0.757 | |
| | Bench press | | 0.739 | |
| | Squat | | 0.664 | |
| | Forehand straight punch force | | 0.644 | |
| | 3-min cumulative punching force | | | 0.810 |
| | Back neck muscle load limit flexion and extension | | | 0.764 |
| | Anterior cervical muscle load limit flexion and extension | | | 0.663 |
| Speed | Backhand straight punch reaction time | 0.957 | | |
| | Forehand straight punch reaction time | 0.938 | | |
| | Backhand straight punch reaction time | | 0.948 | |
| | Forehand straight punch speed | | 0.916 | |
| | 30-m sprint | | | 0.918 |
| | 100-m sprint | | | 0.914 |
| Endurance | 9-min double shake jump rope | 0.848 | | |
| | 12-min sprint | 0.633 | | |
| | 400-m sprint | −0.837 | | |
| Coordination and agility | 1-min double shake jump rope | 0.802 | | |
| | 1-min hexagon jump | 0.746 | | |
| | Eight-way fast slide in boxing ring | −0.506 | | |
| Flexibility | Sitting forward bend | 0.952 | | |
| | Shoulder flexibility (Forehand) | 0.889 | | |
| | Shoulder flexibility (Backhand) | 0.857 | | |

**Table 6 Weighted model of the physical fitness evaluation index for elite male boxers.**

| Primary indexes | Weight | Secondary indexes | | Tertiary indexes | Weight | | Total |
|---|---|---|---|---|---|---|---|
| Body shape | 0.182 | Circumference | | Backhand upper arm circumference differential | 0.317 | | 0.063 |
| | | Length | | Finger span height | 0.283 | | 0.054 |
| | | Plumpness | | Cottrell index | 0.204 | | 0.037 |
| | | Width | | Pelvic width/shoulder width × 100 | 0.197 | | 0.028 |
| Body function | 0.241 | Anaerobic ability | | Relative maximum anaerobic power | 0.321 | | 0.077 |
| | | Aerobic ability | | Relative VO$_{2max}$ | 0.275 | | 0.066 |
| | | Recovery ability | | Creatine kinase | 0.219 | | 0.053 |
| | | Hormone level | | Testosterone | 0.185 | | 0.045 |
| Athletic quality | 0.577 | Strength | Fast strength | Speed strength index | 0.296 | 0.422 | 0.104 |
| | | | Maximum strength | Backhand straight punch force | | 0.407 | 0.040 |
| | | | Strength endurance | 3-min cumulative punching force | | 0.171 | 0.028 |
| | | Speed | Reaction speed | Backhand straight punch reaction time | 0.243 | 0.343 | 0.056 |
| | | | Action speed | Backhand straight punch speed | | 0.334 | 0.047 |
| | | | Movement speed | 30-m sprint | | 0.323 | 0.037 |
| | | Endurance | | 9-min double shake jump rope | 0.191 | | 0.110 |
| | | Coordination and agility | | 1-min double shake jump rope | 0.145 | | 0.083 |
| | | Flexibility | | Sitting forward bend | 0.125 | | 0.072 |

**Table 7 Standards for evaluating the physical fitness level of lightweight elite male boxers.**

| Indexes | | Poor <10% | Fair 10–<25% | Average 25–75% | Good >75– 90% | Excellent >90% |
|---|---|---|---|---|---|---|
| Tertiary indexes | Backhand upper arm circumference differential | <2.72 | 2.72–<3.15 | 3.15–5.30 | >5.30–5.78 | >5.78 |
| | Finger span height | <−0.20 | −0.20–<1.60 | 1.60–3.23 | >3.23–4.10 | >4.10 |
| | Cottrell index | < 301 | 301–<306 | 306–328 | >328–335 | >335 |
| | Pelvic width/shoulder width × 100 | >71.64 | 71.64–>70.37 | 70.37–63.61 | <63.61–61.56 | <61.56 |
| | Relative maximum anaerobic power | <9.37 | 9.37–<10.11 | 10.11–13.09 | >13.09–13.72 | >13.72 |
| | Relative VO$_{2max}$ | <52.72 | 52.72–<55.02 | 55.02–60.00 | >60.00–60.80 | >60.80 |
| | Creatine kinase | >259 | 259–>224 | 224–180 | <180–165 | <165 |
| | Testosterone | <500 | 500–<545 | 545–606 | >606–657 | >657 |
| | Speed strength index | <247 | 247–<261 | 261–306 | >306–334 | >334 |
| | Backhand straight punch force | <225 | 225–<238 | 238–273 | >273–286 | >286 |
| | 3-min cumulative punching force ($10^4$) | <2.973 | 2.973–<3.659 | 3.659–5.050 | >5.050–6.008 | >6.008 |
| | Backhand straight punch reaction time | >0.984 | 0.984–>0.923 | 0.923–0.837 | < 0.837–0.815 | <0.815 |
| | Backhand straight punch speed | <8.15 | 8.15–<8.51 | 8.51–9.68 | >9.68–11.00 | >11.00 |
| | 30-m sprint | >4.69 | 4.69–>4.60 | 4.60–4.26 | <4.26–4.14 | <4.14 |
| | 9-min double shake jump rope | <502 | 502–<561 | 561–718 | >718–756 | >756 |
| | 1-min double shake jump rope | <101 | 101–<110 | 110–128 | >128–132 | >132 |
| | Sitting forward bend | <8.98 | 8.98– <12.61 | 12.61–17.51 | >17.51–20.71 | >20.71 |

| Indexes | | Poor <10% | Fair 10–<25% | Average 25–75% | Good >75–90% | Excellent >90% |
|---|---|---|---|---|---|---|
| Primary indexes | Body shape | <6.10 | 6.10–<7.78 | 7.78–11.48 | >11.48–13.80 | >13.80 |
| | Body function | <4.12 | 4.12–<6.64 | 6.64–12.48 | >12.48–14.98 | >14.98 |
| Athletic quality | Strength | <2.91 | 2.91–<6.07 | 6.07–12.83 | >12.83–16.55 | >16.55 |
| | Speed | <5.15 | 5.15–<7.32 | 7.32–11.87 | >11.87–14.86 | >14.86 |
| | Athletic quality (Composite) | <5.46 | 5.46–<7.11 | 7.11–13.02 | >13.02–14.74 | >14.74 |
| Composite indicators | | <6.16 | 6.16–<7.46 | 7.46–12.66 | >12.66–14.72 | >14.72 |

**Table 8 Standards for evaluating the physical fitness level of middleweight elite male boxers.**

| Indexes | | Poor <10% | Fair 10–<25% | Average 25–75% | Good >75–90% | Excellent >90% |
|---|---|---|---|---|---|---|
| Tertiary indexes | Backhand upper arm circumference differential | <2.72 | 2.72–<3.75 | 3.75–5.30 | >5.30–6.51 | >6.51 |
| | Finger span height | <−0.20 | −0.20–<2.10 | 2.10–3.50 | >3.50–4.23 | >4.23 |
| | Cottrell index | <301 | 301–<358 | 358–403 | >403–421 | >421 |
| | Pelvic width/shoulder width × 100 | >75.95 | 75.95–>72.03 | 72.03–65.52 | <65.52–61.62 | <61.62 |
| | Relative maximum anaerobic power | <8.34 | 8.34–<9.32 | 9.32–12.87 | >12.87–13.12 | >13.12 |
| | Relative VO$_{2max}$ | <52.74 | 52.74–<54.95 | 54.95–59.52 | >59.52–61.06 | >61.06 |
| | Creatine kinase | >254 | 254–>232 | 232–194 | <194–181 | <181 |
| | Testosterone | <521 | 521–<597 | 597–704 | >704–786 | >786 |
| | Speed strength index | <267 | 267–<303 | 303–379 | >379–403 | >403 |
| | Backhand straight punch force | <261 | 261–<282 | 282–333 | >333–347 | >347 |
| | 3-min cumulative punching force ($10^4$) | <3.396 | 3.396–<4.206 | 4.206–5.321 | >5.321–5.898 | >5.898 |
| | Backhand straight punch reaction time | >1.028 | 1.028–>0.960 | 0.960–0.860 | <0.860–0.804 | <0.804 |
| | Backhand straight punch speed | <8.31 | 8.31–<8.76 | 8.76–10.42 | >10.42–11.08 | >11.08 |
| | 30-m sprint | >4.66 | 4.66–>4.51 | 4.51–4.27 | <4.27–4.12 | <4.12 |
| | 9-min double shake jump rope | <490 | 490–<523 | 523–637 | >637–738 | >738 |
| | 1-min double shake jump rope | <98 | 98–<107 | 107–125 | >125–135 | >135 |
| | Sitting forward bend | <8.53 | 8.53–<13.41 | 13.41–19.19 | >19.19–23.33 | >23.33 |
| Primary indexes | Body shape | <6.34 | 6.34–<7.40 | 7.40–12.38 | >12.38–13.60 | >13.60 |
| | Body function | <4.20 | 4.20–<6.91 | 6.91–12.74 | >12.74–15.07 | >15.07 |
| Athletic quality | Strength | <2.73 | 2.73–<5.61 | 5.61–13.81 | >13.81–16.57 | >16.57 |
| | Speed | <4.43 | 4.43–<6.66 | 6.66–12.66 | >12.66–14.94 | >14.94 |
| | Athletic quality (Composite) | <5.50 | 5.50–<8.22 | 8.22–12.28 | >12.28–14.56 | >14.56 |
| Composite indicators | | <6.25 | 6.25–<8.09 | 8.09–12.06 | >12.06–13.52 | >13.52 |

including body shape, physical function, and athletic ability to determine the contribution rate of each first-level indicator.

## Establishment of the model

Physical fitness level evaluation standards were established for the lightweight (Table 7), middleweight (Table 8), and heavyweight categories (Table 9). An athlete's physical fitness

**Table 9 Standards for evaluating the physical fitness level of heavyweight elite male boxers.**

| Indexes | | Poor <10% | Fair 10–< 25% | Average 25–75% | Good >75–90% | Excellent >90% |
|---|---|---|---|---|---|---|
| Tertiary indexes | Backhand upper arm circumference differential | <3.89 | 3.89–<4.33 | 4.33–5.70 | >5.70–6.19 | >6.19 |
| | Finger span height | 2.35 | 2.35–<3.30 | 3.30–6.55 | >6.55–7.28 | >7.28 |
| | Cottrell index | <430 | 430–<441 | 441–499 | > 499–556 | >556 |
| | Pelvic width/shoulder width × 100 | >79.95 | 79.95–>79.04 | 79.04–70.99 | <70.99–67.56 | <67.56 |
| | Relative maximum anaerobic power | <7.77 | 7.77–<8.98 | 8.98–11.78 | >11.78–12.46 | >12.46 |
| | Relative $VO_{2max}$ | <44.78 | 44.78–<49.28 | 49.28–54.01 | >54.01–56.52 | >56.52 |
| | Creatine kinase | >286 | 286–>245 | 245–196 | < 196–188 | <188 |
| | Testosterone | <455 | 455–<507 | 507–603 | >603–692 | >692 |
| | Speed strength index | <267 | 267–<303 | 303–379 | >379–403 | >403 |
| | Backhand straight punch force | <281 | 281–<329 | 329–370 | >370–418 | >418 |
| | 3-min cumulative punching force ($10^4$) | <3.882 | 3.882–<4.794 | 4.794–5.888 | >5.888–6.841 | >6.841 |
| | Backhand straight punch reaction time | >1.025 | 1.025–>0.983 | 0.983–0.893 | <0.893–0.855 | <0.855 |
| | Backhand straight punch speed | <8.22 | 8.22–<8.71 | 8.71–9.56 | >9.56–10.51 | >10.51 |
| | 30-m sprint | >4.87 | 4.87–>4.66 | 4.66–4.33 | <4.33–4.27 | <4.27 |
| | 9-min double shake jump rope | <455 | 455–<499 | 499–578 | >578–622 | >622 |
| | 1-min double shake jump rope | <76 | 76–<91 | 91–116 | >116–120 | >120 |
| | Sitting forward bend | <9.20 | 9.20–<13.43 | 13.43–23.41 | >23.41–26.25 | >26.25 |
| Primary indexes | Body shape | <5.28 | 5.28–<6.33 | 6.33–12.26 | >12.26–13.95 | >13.95 |
| | Body function | <4.71 | 4.71–<7.14 | 7.14–12.91 | >12.91–14.75 | >14.75 |
| Athletic quality | Strength | <2.05 | 2.05–<5.42 | 5.42–14.10 | >14.10–17.20 | >17.20 |
| | Speed | <4.03 | 4.03–<6.61 | 6.61–13.18 | >13.18–15.64 | >15.64 |
| | Athletic quality (Composite) | <5.09 | 5.09–<6.82 | 6.82–12.28 | >12.28–13.67 | >13.67 |
| Composite indicators | | <5.59 | 5.59–<7.20 | 7.20–11.41 | >11.41–13.44 | >13.44 |

development requires long and complex training (*Ambrozy et al., 2021*). Therefore, a clear long-term goal for a boxer must be established. Taking the 90th percentile as the baseline for an ideal physical fitness model, the values of various physical fitness indicators were calculated for elite male boxers. The results were compiled to establish an ideal model for physical fitness for elite Chinese male boxers in the lightweight, middleweight, and heavyweight categories (Table 10).

To assess the accuracy of the evaluation standards in this physical fitness index a regression analysis was conducted on all measured data. First, the athlete's original testing data was compiled and converted into corresponding scores for body shape, body function, athletic ability, and comprehensive physical fitness. These scores were based on the evaluation standards and weighted factors of each index. Frequency statistics were calculated using the rating standards (Table 11) and the results were subjected to multiple chi-squared tests. A significant difference ($p < 0.001$) between elite and first-class athletes was observed for body shape, body function, athletic ability, and comprehensive physical fitness scores. This strongly indicates that the evaluation standards in these four areas

**Table 10 Ideal model of the physical structure characteristics of elite male boxers.**

| Physical structure | Indexes | Unit | Ideal model value ≥ 90% | | |
|---|---|---|---|---|---|
| | | | Lightweight | Middleweight | Heavyweight |
| Body shape | Backhand upper arm circumference differential | cm | ≥5.78 | ≥6.51 | ≥6.19 |
| | Finger span height | cm | ≥4.10 | ≥4.23 | ≥7.28 |
| | Cottrell index | kg/cm | ≥335 | ≥421 | ≥556 |
| | Pelvic width/shoulder width × 100 | cm/cm | ≤61.56 | ≤61.62 | ≤67.56 |
| Body function | Relative maximum anaerobic power | w/kg | ≥13.72 | ≥13.12 | ≥12.46 |
| | Relative VO$_{2max}$ | ml/kg/min | ≥60.80 | ≥61.06 | ≥56.52 |
| | Creatine kinase | u/l | ≤165 | ≤181 | ≤188 |
| | Testosterone | ng/dl | ≥657 | ≥786 | ≥692 |
| Athletic quality | Speed strength index | kg/s | ≥334 | ≥403 | ≥469 |
| | Backhand straight punch force | kg | ≥286 | ≥347 | ≥418 |
| | 3-min cumulative punching force | $10^4$kg | ≥6.008 | ≥5.898 | ≥6.841 |
| | Backhand straight punch reaction time | s | ≤0.815 | ≤0.804 | ≤0.855 |
| | Backhand straight punch speed | m/s | ≥11.00 | ≥11.08 | ≥10.51 |
| | 30-m sprint | s | ≤4.14 | ≤4.12 | ≤4.27 |
| | 9-min double shake jump rope | pcs | ≥756 | ≥738 | ≥622 |
| | 1-min double shake jump rope | pcs | ≥132 | ≥135 | ≥120 |
| | Sitting forward bend | cm | ≥20.71 | ≥23.33 | ≥26.25 |

**Table 11 Frequency statistics for the evaluation criterion of elite male boxer.**

| Grade | Elite | | | | First-class | | | |
|---|---|---|---|---|---|---|---|---|
| | Body shape | Body function | Athletic quality | Composite | Body shape | Body function | Athletic quality | Composite |
| Excellent | 16 | 14 | 14 | 15 | 7 | 1 | 1 | 0 |
| Good | 27 | 25 | 29 | 30 | 10 | 5 | 5 | 2 |
| Average | 27 | 32 | 32 | 30 | 21 | 24 | 23 | 26 |
| Fair | 4 | 4 | 1 | 1 | 22 | 27 | 30 | 29 |
| Poor | 2 | 1 | 0 | 0 | 13 | 16 | 14 | 16 |
| Total | 76 | 76 | 76 | 76 | 73 | 73 | 73 | 73 |

effectively reflect the actual level of an athlete's physical fitness. These standards can be used to assess the physical fitness levels of outstanding male boxers.

## Assessment of physical fitness development levels

Based on the physical fitness test results and the corresponding index scores of 149 elite male boxers, critical values (mean ± standard deviation (SD)) of the best and worst of all the indexes were determined for boxers in the lightweight, middleweight, and heavyweight categories (Table 12). Parameters higher than the critical value (mean + SD) were considered superior indexes, whereas those lower than the critical value (mean − SD) were considered inferior.

**Table 12 Physical fitness indexes' critical values in three categories using radar analysis.**

| Indexes | Lightweight | | Middleweight | | Heavyweight | |
|---|---|---|---|---|---|---|
| | Worst value M−SD | Best value M+SD | Worst value M−SD | Best value M+SD | Worst value M−SD | Best value M+SD |
| Backhand upper arm circumference differential | 4.09 | 15.42 | 4.60 | 15.40 | 4.16 | 15.79 |
| Finger span height | 4.23 | 15.87 | 4.53 | 15.38 | 3.72 | 15.48 |
| Cottrell index | 4.05 | 15.66 | 5.09 | 15.29 | 3.98 | 15.42 |
| Pelvic width/shoulder width × 100 | 3.73 | 15.35 | 4.11 | 15.24 | 3.73 | 15.47 |
| Relative maximum anaerobic power | 3.73 | 15.35 | 3.77 | 15.34 | 3.81 | 15.44 |
| Relative VO$_{2max}$ | 3.77 | 15.60 | 3.88 | 15.45 | 3.71 | 15.44 |
| Creatine kinase | 3.73 | 15.39 | 4.00 | 15.56 | 3.89 | 15.51 |
| Testosterone | 3.83 | 15.49 | 3.89 | 15.40 | 3.80 | 15.50 |
| Speed strength index | 3.87 | 15.49 | 3.85 | 15.33 | 3.76 | 15.40 |
| Backhand straight punch force | 3.84 | 15.43 | 3.86 | 15.43 | 3.81 | 15.49 |
| 3-min cumulative punching force (10$^4$) | 3.79 | 15.34 | 3.73 | 15.27 | 3.65 | 15.45 |
| Backhand straight punch reaction time | 3.82 | 15.50 | 3.83 | 15.38 | 3.92 | 15.48 |
| Backhand straight punch speed | 4.05 | 15.61 | 4.06 | 15.59 | 4.34 | 15.56 |
| 30-m sprint | 4.08 | 15.48 | 3.89 | 15.55 | 4.14 | 15.46 |
| 9-min double shake jump rope | 3.81 | 15.31 | 4.83 | 15.99 | 3.71 | 15.39 |
| 1-min double shake jump rope | 4.12 | 15.78 | 9.66 | 19.10 | 4.05 | 15.65 |
| Sitting forward bend | 3.74 | 15.38 | 2.77 | 11.96 | 3.71 | 15.44 |

**Table 13 Three categories of the challenge goal fitness model for elite male boxers.**

| Physical structure | Indexes | Unit | Lightweight | Middleweight | Heavyweight |
|---|---|---|---|---|---|
| Body shape | Backhand upper arm circumference differential | cm | 6.30 | 7.10 | 6.60 |
| | Finger span height | cm | 5.00 | 9.20 | 10.20 |
| | Cottrell index | kg/cm | 339 | 430 | 621 |
| | Pelvic width/shoulder width × 100 | cm/cm | 56.38 | 57.240 | 64.95 |
| Body function | Relative maximum anaerobic power | w/kg | 14.73 | 13.65 | 12.86 |
| | Relative VO$_{2max}$ | ml/kg/min | 62.51 | 63.13 | 60.36 |
| | Creatine kinase | u/l | 104 | 114 | 125 |
| | Testosterone | ng/dl | 792 | 924 | 905 |
| Athletic quality | Speed strength index | kg/s | 403 | 471 | 483 |
| | Backhand straight punch force | kg | 326 | 355 | 445 |
| | 3-min cumulative punching force | 10$^4$kg | 7.596 | 7.224 | 8.287 |
| | Backhand straight punch reaction time | second | 0.785 | 0.703 | 0.847 |
| | Backhand straight punch speed | m/s | 11.72 | 12.17 | 11.49 |
| | 30-m sprint | second | 4.07 | 3.98 | 4.11 |
| | 9-min double shake jump rope | pcs | 800 | 850 | 875 |
| | 1-min double shake jump rope | pcs | 155 | 140 | 132 |
| | Sitting forward bend | cm | 24.47 | 35.78 | 34.56 |

During the second stage of our study, radar analysis was conducted to diagnose the individual physical fitness level of 25 male boxers with a dedicated chart prepared for each athlete. The radar analysis chart clearly illustrates the first-level physical fitness indicators and their respective categories.

Based on the optimal values of the athlete fitness indicators in the first stage, a challenge-goal fitness model for all three weight categories was created for elite male boxers (Table 13). The difference coefficient was calculated for all indicators based on the 25 athletes during the second stage and the weighted factors for each level were analysed as well as the mean of all the difference coefficients at various levels. The assessment results showed a significant overall difference in physical function and athletic ability across the three weight categories. Flexibility was suboptimal for all athletes. Athletes in the middleweight group needed to improve their punching distance, load adaptation, recovery ability, hormone levels, and flexibility. Meanwhile, athletes in the heavyweight group had relatively low levels of endurance and flexibility. Attention must be paid to the issues common to all weight groups in order to enhance athletes' fitness levels, both for targeted fitness levels as well as overall improvement.

## DISCUSSION

### Analysis of physical fitness evaluation system indicators

Boxing is a combat sport primarily focused on delivering upper-limb strikes. Boxers with longer arms have an advantage in terms of their punching distance (Han et al., 2020). Boxers with greater body control have improved defensive capabilities and those with a low body fat percentage are often outstanding in this field (Kim, Song & Min, 2016). The backhand upper arm circumference differential serves as an indirect indicator of muscle contraction strength and explosive power. Larger differences indicate better muscle quality (Chottidao et al., 2022). The finger span height ratio reflects a boxer's three-dimensional space range, which is crucial for both offensive and defensive maneuvers. The Cottrell index assesses body proportions and muscle mass and plays a role in evaluating upper body development and flexibility (Chaabene et al., 2015). A pelvic width/shoulder width × 100 is an important morphological indicator of trunk shape and can reflect the strength of the upper limbs and shoulder girdle. Athletes with lower values tend to be more flexible.

In addition to body shape, athletic ability is closely related to body function, which is one of the most important factors in athletic performance (Hukkanen & Hakkinen, 2017). Fitness training can not only change the structure of the body but also modify and activate its corresponding physiological and biochemical functions. This results in adaptive changes that enhance athletic performance (Hanon, Savarino & Thomas, 2015; Slimani et al., 2017). Research has shown that amateur boxing matches are intermittent in nature. They are characterized by short bouts that have high workloads and that require athletes to demonstrate intense and explosive power at irregular intervals (Loturco et al., 2018; Siegler & Hirscher, 2010). Athletes must have good anaerobic and aerobic capabilities, which not only enhances their ability to punch with increased strength and speed but also helps them maintain a stable power output and recover energy during rest intervals (Chamari &

*Padulo, 2015*; *Nassib et al., 2017*). Relative maximum anaerobic power and relative $VO_{2max}$ were measured to assess how well boxing athletes were able to maintain and recover energy. Creatine kinase and testosterone are biochemical indicators of how the body is functioning. After high-intensity training, creatine kinase levels significantly increase and changes in creatine kinase activity post-training can reflect the athlete's level of recovery (*Ehlers, Ball & Liston, 2002*). Serum testosterone level is closely related to an athlete's maximum strength and speed; generally, the higher their basal level of serum testosterone, the better their ability to withstand and recover from high-load training. This helps boxers adapt to greater training loads (*Obminski, Borkowski & Sikorski, 2011*).

Boxing training should focus on developing an athlete's comprehensive fitness levels. Tailored training plans are based on the condition of the athlete and the characteristics of the sport with an emphasis on the development of athlete-specific skills (*Davis, 2017*; *Piorkowski, Lees & Barton, 2011*). The ability to punch effectively requires advanced muscle strength and explosive power. Elite boxers must move very quickly the moment they have an offensive, defensive, or counter-attack opportunity in order to rapidly punch and accurately hit the opponent's vulnerable areas and seize an advantageous position (*Dunn et al., 2022*; *Han et al., 2020*; *Hukkanen & Hakkinen, 2017*; *Lenetsky et al., 2020*).

According to the physical fitness classification, a boxer wins by effectively hitting parts of their opponent with their fists. Strength training is the foundation for maintaining and improving their competitive skills (*Beattie & Ruddock, 2022*; *Bingül et al., 2018*; *Lenetsky, Harris & Brughelli, 2013*). The speed strength index, which measures the ability of a boxer to hit objects with maximum force in the shortest possible time, is a test indicator of upper-limb explosive power. The strength of the backhand straight punch is a direct manifestation of the athlete's maximum strength in a boxing competition. It is used to evaluate the specific strength of an athlete (*Smith et al., 2000*; *Yi et al., 2022*). The 3-min cumulative punching force test requires boxers to punch with maximum strength and each punch must have a certain speed and power. It is performed to assess an athlete's anaerobic capacity and muscle endurance (*Dunn et al., 2022*). Speed quality consists of reaction speed, action speed, and movement speed. Excellent reaction and punching speeds are required to ensure that athletes successfully punch the opponent's vulnerable areas when seizing an opportunity to attack or counterattack during a bout (*Hukkanen & Hakkinen, 2017*; *López-Laval et al., 2020*; *Stanley et al., 2018*). A straight punch with a rear cross is one of the most effective punches because of its high striking force (*Lenetsky et al., 2020*; *Yi et al., 2022*). The backhand straight punch reaction time and punch speed are used to assess an athlete's reaction and action speeds. The 30-m sprint can be used to evaluate an athlete's movement speed, as it requires very similar footwork and foot speed as that demonstrated in boxing (*Cepulenas et al., 2011*; *El-Ashker, 2018*).

The 9-min double shake jump rope swing can be a good indicator of muscle endurance, as it combines both strength and endurance and matches well with the 3- × 3-min time allocation in boxing. The 1-min double shake jump rope task is one of the training methods most used to enhance coordination and agility. Good coordination and agility enable athletes to complete highly difficult movements, be more efficient in their work, and conserve their energy. This allows them to have a competitive fitness edge during intense

combat (*Nimphius et al., 2018*). Finally, excellent flexibility allows boxers to use challenging offensive and defensive techniques during bouts. The sitting forward bend test mainly reflects the shoulder, chest, waist, hip, and knee joint flexibility of athletes and provides a comprehensive evaluation of flexibility (*El-Ashker, 2018*).

### Validity of the findings

Athletes' physical fitness status is highly dynamic as it is affected by many factors (*Halson, 2014*). It is necessary to assess the physical fitness level of boxers in order to develop and improve their athletic competition skills and to help them transition from their current boxing level to a higher level (*Huang et al., 2021*).

Physical fitness evaluation standards for male boxing athletes in China were established using the percentile method and a 20-point scoring system. The 90th percentile of physical fitness indicator values was used as the baseline to build an ideal model. The objectivity and effectiveness of the evaluation was shown using regression analyses. Assessments of an athlete's boxing status can reflect their actual level of physical fitness, which indicates that the critical values used and the proposed challenge-goal models developed are scientifically effective. It is recommended that these models are applied in training practice.

### Limitations and future outlook

There were some limitations in our study. First, our small sample size and the vast differences among athletes mean the evaluation criteria may not be universally applicable. It is necessary in future studies to recruit more athletes to further develop the evaluation criteria. Second, the evaluation criteria were not formulated based on all eight weight classes. Further refinement of the physical fitness evaluation criteria by weight class may help assess the physical fitness levels of boxers in each weight class more accurately. At the same time, when using evaluation criteria, the inherent interaction between various components of physical fitness should also be considered, such as the interaction between the explosive force and speed indicators. In addition, the content and standards of the physical fitness evaluation system should be revised and improved when any future changes in the rules of amateur boxing occur.

## CONCLUSION

We recommend the use of a three-level evaluation system for the physical fitness assessment of male boxing athletes. The three levels are: (1) physical form indicators including backhand upper arm circumference relaxation difference, finger span height, Cottrell index, and iliac width/shoulder width × 100; (2) physical function indicators including relative maximum anaerobic power, relative maximum oxygen uptake, creatine kinase, and testosterone levels; and (3) athletic ability indicators including the fast strength index, backhand straight punch hitting power, 3-min cumulative hitting power, backhand straight punch reaction time, backhand straight punch speed, 30-m sprint, 9-min double-arm swing, 1-min double-arm swing, and sitting forward bend.

## ACKNOWLEDGEMENTS

We thank all the boxers, coaches, and researchers for their voluntary participation in this study.

### Funding

The research was funded by the Scientific and Technological Services for Physical Fitness Training of Key Athletes of the Chinese Boxing Team in 2022, China, Project ID: No. 2022AY010. The funders had no role in study design, data collection and analysis, decision to publish, or preparation of the manuscript.

### Grant Disclosures

The following grant information was disclosed by the authors:
Scientific and Technological Services for Physical Fitness Training of Key Athletes of the Chinese Boxing Team in 2022, China: 2022AY010.

### Competing Interests

The authors declare that they have no competing interests.

### Author Contributions

- Guodong Wu conceived and designed the experiments, performed the experiments, analyzed the data, prepared figures and/or tables, authored or reviewed drafts of the article, and approved the final draft.
- Yuqiang Guo performed the experiments, analyzed the data, prepared figures and/or tables, authored or reviewed drafts of the article, and approved the final draft.
- Liqin Zhang conceived and designed the experiments, analyzed the data, authored or reviewed drafts of the article, and approved the final draft.
- Chao Chen conceived and designed the experiments, performed the experiments, analyzed the data, prepared figures and/or tables, authored or reviewed drafts of the article, and approved the final draft.

### Human Ethics

The following information was supplied relating to ethical approvals (*i.e.*, approving body and any reference numbers):

The amateur boxers volunteered to participate in this study, which was approved by the Shanghai University of Sport Research Ethics Committee and in accordance with the Helsinki declaration.

### Data Availability

The data is available at figshare: Guo, Yuqiang (2023). The boxers' data of physical fitness. figshare. Dataset. https://doi.org/10.6084/m9.figshare.22306735.v4.

## Supplemental Information

Supplemental information for this article can be found online at http://dx.doi.org/10.7717/peerj.17271#supplemental-information.

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
