# Peer review of "A physical fitness–evaluation system for outstanding Chinese male boxers"

_PeerJ, doi:10.7717/peerj.17271_

## Round 0.1 · original submission · Minor Revisions

Reviewer 2 has raised some important questions. Please respond to questions and comments from both reviewers with new line numbers to indicate where changes have been made.

Reviewer 1 ·

Basic reporting

The manuscript is of great importance for the area in which the authors carry out their experimental research. Bringing important information to the area of ​​self-performance sports and combat sports and their evaluations, which in the literature present few studies. The introduction is well written but I suggest that the authors carry out a literature review including more recent research for the points they address in the introduction and can also look at physiological and biochemical analyzes to determine the athlete's performance.
Zhong H, Bu X. Development of a Sensitive Quality Evaluation System for Chinese Outstanding Female Boxers. J Environ Public Health. 2022 Jul 30;2022:4664938. doi: 10.1155/2022/4664938. PMID: 35942146; PMCID: PMC9356856.
Zhang G, Chen X, Xiao L, Li Y, Li B, Yan Z, Guo L, Rost DH. The Relationship Between Big Five and Self-Control in Boxers: A Mediating Model. Front Psychol. 2019 Aug 8;10:1690. doi: 10.3389/fpsyg.2019.01690. PMID: 31440177; PMCID: PMC6694765.
Merlo R, Rodríguez-Chávez Á, Gómez-Castañeda PE, Rojas-Jaramillo A, Petro JL, Kreider RB, Bonilla DA. Profiling the Physical Performance of Young Boxers with Unsupervised Machine Learning: A Cross-Sectional Study. Sports (Basel). 2023 Jul 7;11(7):131. doi: 10.3390/sports11070131. PMID: 37505618; PMCID: PMC10384265.
Kılıc Y, Cetin HN, Sumlu E, Pektas MB, Koca HB, Akar F. Effects of Boxing Matches on Metabolic, Hormonal, and Inflammatory Parameters in Male Elite Boxers. Medicina (Kaunas). 2019 Jun 18;55(6):288. doi: 10.3390/medicina55060288. PMID: 31216765; PMCID: PMC6630693.

Experimental design

no comment

Validity of the findings

no comment

Additional comments

Congratulations to the researchers for research that has little scope in academia. I suggest that researchers take a broader look at physiological, biochemical, immunological analyzes and electromygographic parameters.

Reviewer 2 ·

Basic reporting

The english language is confusing or ambiguous at times, especially in the introduction. Here are some examples:
"..and an ideal value model was proposed. " difficult to understand
"The 10-Point Must System brings boxing into combat." lacking context and therefore hard to understand
“Comprehensive physical fitness is the foundation of boxers’ physical conditioning emphasizes training ...” invalid sentence
“The participants were volunteered to participate in this study” sounds like participants were forced into participation, "The participants volunteered to participate" would be better

The authors attempt to provide background for their study, but the background for their statistical analyses is usually lacking.

The article structure and tables are mostly clear. The authors did not provide figures, which could have aided in presentation in some cases. I feel that table 1 would have better presentation in a figure format. Without legends, the table titles are sometimes hard to understand. Like in table 1 "Training age" is ambiguous, is this age the athletes started training of the number of years in training?

Experimental design

When conducting statistical analyses on the data, it would be better to explain the rationale behind the usage of certain tests and a basic explanation of the tests performed would be beneficial. For example, when mentioning factor load analysis, please explain that it is a correlation test, explain what you actually testing (correlation between X and Y) and describe what is the common threshold used as cutoff (and what kind of numbers are seen for your analysis). As another example, why did you use Kaiser orthogonal rotation, and the KMO and Bartlett’s tests, and what do the results actually indicate?

Formula (1) is not valid as written: in Formulas (1) and the text following, the subscript of b is labeled as y, but the sum is calculated over j. Additionally, k and i are not explained.
The connection between Formula (1) and (2) is not obvious.

The results in table 2 indicate that most testing indexes had a significance score of 4.2-4.8 on a 5 point scale. This seems concerning as the basically all indexes were weighted equally and the full range was not used. More explanation would help. Would a wider set of indexes have helped? Or a different point system?

Validity of the findings

The measurements from athletes were shared but data behind table 2 was not.

Interactions between indexes should be considered.

If the system is adopted by new users, is the system robust if not all the measurements can be obtained?

---

## Round 0.2 · Minor Revisions

I have one final suggestion. Please change the sentence starting on line 44. It should read,

>In 2013, a major change in how a boxing contest could be won was made: The International Boxing Federation replaced the "point-to-win" system with the “10-point must" system. Under the new rules, each round is scored using four levels:

Reviewer 2 ·

Basic reporting

no comment

Experimental design

no comment

Validity of the findings

no comment

Additional comments

The authors have addressed my comments sufficiently through the rebuttal. I have no remaining concerns.

---

## Round 0.3 · accepted · Accept

The authors have addressed all recommendations and queries from reviewers and the manuscript is now ready for publication.